# TVLT: Textless Vision-Language Transformer

**Zineng Tang**[*]   **Jaemin Cho**[*]   **Yixin Nie**[*]   **Mohit Bansal**
UNC Chapel Hill
{terran, jmincho, yixin1, mbansal}@cs.unc.edu

## Abstract

In this work, we present the Textless Vision-Language Transformer (TVLT), where homogeneous transformer blocks take raw visual and audio inputs for vision-and-language representation learning with minimal modality-specific design, and do not use text-specific modules such as tokenization or automatic speech recognition (ASR). TVLT is trained by reconstructing masked patches of continuous video frames and audio spectrograms (masked autoencoding) and contrastive modeling to align video and audio. TVLT attains performance comparable to its text-based counterpart on various multimodal tasks, such as visual question answering, image retrieval, video retrieval, and multimodal sentiment analysis, with 28x faster inference speed and only 1/3 of the parameters. Our findings suggest the possibility of learning compact and efficient visual-linguistic representations from low-level visual and audio signals without assuming the prior existence of text.[1]

## 1 Introduction

Humans perceive and learn the external world through signals from multiple modalities. To embody such human learning in machines, substantial research efforts are dedicated to developing vision-and-language (VL) models that can understand the joint semantics between visual and linguistic modalities and solve tasks such as visual question answering [4]. Although most such VL models use written language rather than spoken language as the main verbal communication channel, the default communication modality among humans has been speech, since circa 100,000 BCE [77]. Written language is relatively recent; cuneiform script, the earliest writing system, was developed circa 3,200 BCE [65]. Moreover, we have witnessed an increasing usage of AI models in real-world products such as virtual assistants and smart speakers [40], where perception-level signals such as video and audio are the natural form of input. Intuitively, direct modeling of such signals will potentially yield more compact and efficient representations.

Transformers [80] have recently achieved great success in vision-language representation learning [75; 10; 48; 73; 86; 85] by using text-based modules [15] on text-annotated images or videos. However, it is non-trivial to learn VL representations using transformers that take only low-level visual and acoustic inputs without the prior existence of written language. The challenge lies in the difference between text and acoustic signals; text is discrete and dense in information, while acoustic signals are continuous and sparse in information [26; 7]. Therefore, modality-specific architectures have been used to model data from different modalities. It is only recently that researchers started using modality-agnostic transformer architecture to learn representations of different unimodal [17; 19; 8], vision-text [32; 54], or vision-audio-text [2] data. However, to the best of our knowledge, no previous work has explored a single homogeneous (modality-agnostic) minimalist transformer that learns visual-linguistic representations directly from visual and acoustic input at the perception level (without relying on text), and also makes the textless VL model more compact and efficient than the existing text-based VL models (see Sec. 2 for details).

---

[*]equal contribution
[1]Our code and checkpoints are available at: https://github.com/zinengtang/TVLT

36th Conference on Neural Information Processing Systems (NeurIPS 2022).

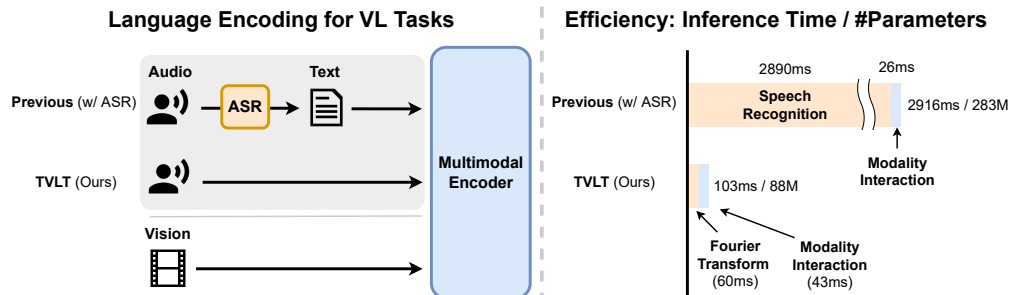

Figure 1: Comparison of previous VL architectures and our proposed textless framework TVLT. The removal of automatic speech recognition (ASR) from the VL pipeline brings efficiency improvement while maintains competitive performance. For inference time calculation, we use 8 video frames and 20s audio (see Sec. 6.2 for detail). As shown in Table 1, TVLT achieves competitive performance to text-counterpart on video retrieval and multimodal sentiment analysis tasks.

In this work, we propose Textless Vision-Language Transformer (TVLT) for vision-and-language representation learning based on video data as the natural source of raw visual and audio input. As depicted in Fig. 2, TVLT accepts low-level video frames and audio spectrograms as input. We employ a minimalist design for TVLT where homogeneous transformer blocks are used for both the encoder and decoder. TVLT is trained by reconstructing masked patches of continuous video frames and audio spectrograms (masked autoencoding) and contrastive modeling to align video and audio. More importantly, TVLT makes no assumptions about the existence of written language and does not involve explicit modeling of text input, such as automatic speech recognition (ASR) or tokenization, which are crucial submodules in the success of existing VL models in aligning written concepts with visual clues.

Despite the removal of text-based modules and modality-specific designs, TVLT achieves results comparable to its text-based counterparts in multimodal tasks (with either direct audio input, or text converted to audio input via TTS) such as visual question answering, image retrieval, video retrieval, and multimodal sentiment analysis, while being computationally efficient with 1/3 parameters and a 28x faster inference speed, as illustrated in Fig. 1. This indicates that the removal of text-specific modules such as ASR in vision-and-language modeling helps reduce computational redundancy in existing pipelined learning paradigms, where text is first extracted through ASR and then further processed by a text-based VL model. Furthermore, we also show that TVLT can capture acoustic information beyond speech and is more effective in multimodal emotion classification than its text-based counterpart. We hope that our findings spark further research in the realm of textless VL models that take raw signals as input and seek to learn a more compact and efficient vision-and-language representation.

## 2   Related Work

**Text-based Representation Learning.**   Large-scale unsupervised pretraining of contextualized language models based on written texts has seen great success in recent years. ELMo [58] proposes to pretrain and finetune a large recurrent language model, which improves performance on a diverse set of downstream natural language processing tasks. BERT [15] improves the scalability of the pretrain-then-finetune paradigm by using a transformer [80] model with a masked language modeling objective. Since then, the pre-training of transformers has been extensively explored for transfer learning in language [46; 82; 38; 16; 72; 60; 13]. In these methods, learning is focused on eliciting high-level linguistic semantics and structures from unlabeled written texts or natural sequences of words.

**Audio-based Representation Learning.**   Pretraining methods on audio input involve transferring the continuous 1D audio signal into dense vectors that can be input to a speech or acoustic model. Early work mainly uses recurrent neural networks [12; 11; 69] and convolution networks [66] for audio encoding. To take advantage of the proven expressiveness and genericity of transformers, more recent work proposed using audio spectrograms [19; 20; 7] as image input and then encoding the patches of such images with a transformer, following the same methodology in computer vision [17].

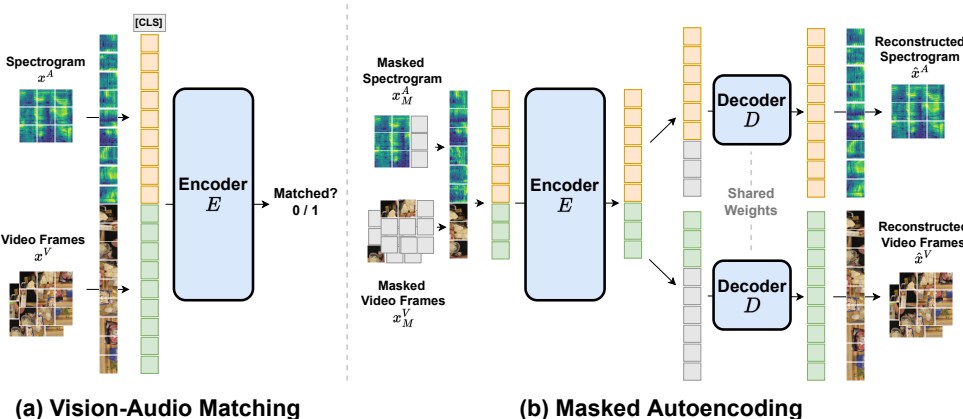

Figure 2: TVLT is pretrained with two objectives: (a) vision-audio matching (Sec. 4.1) and (b) masked autoencoding (Sec. 4.2). The model takes video frames and audio spectrogram as inputs and does not use text input and completely removes text from the pipeline.

The pretraining objectives for transformers range from classification [19] to masked audio modeling [20; 7]. A line of work uses an audio transformer with discrete audio units for pretraining [27] and speech tasks such as generative spoken language modeling [37; 31] and speech emotion conversion [35]. These works focus on learning the acoustic and linguistic characteristics of a language from raw audio or spectrogram.

**Vision-and-Language Representation Learning.** Following the success of pretraining of transformer language models, pretraining of image+text [75; 48; 10; 43; 89; 41], video+text [73; 52; 91; 51; 42; 76; 86], and video+text+audio [78; 84; 61; 85; 2] multimodal transformers has recently achieved improvements in downstream VL tasks such as visual question answering [4; 28] and text-to-video retrieval [81; 90]. These methods use text, such as written captions or ASR transcripts, as input into the language channel. There is another line of work on models taking video+audio input, where they can utilize naturally synchronized vision+audio pairs from videos. Audio-visual synchronization is often used for self-supervised learning [56; 5; 55; 34; 6; 53; 49], or for downstream tasks such as automatic speech recognition [1; 71; 70] and video retrieval [74; 63; 64; 45]. Our work is different from these works, in that we focus on the design of a homogeneous and modality-agnostic transformer (Sec. 3) to achieve a novel, unified, and minimalist textless visual-linguistic representation learning method directly from visual and acoustic signals (without relying on text), via masked autoencoding and contrastive modeling objectives (Sec. 4), which also makes the textless VL model more compact and efficient than the existing text-based VL models.

## 3 TVLT: Textless Vision-Language Transformer

We introduce TVLT: **T**extless **V**ision-**L**anguage **T**ransformer, a minimal end-to-end vision-and-language transformer model that accepts a list of embeddings obtained directly from perception-level video and audio input *without text-specific modules*, as depicted in Fig. 1 and Fig. 2.

### 3.1 Input Embeddings

The input embeddings of TVLT are the sum of (1) modality embedding, (2) temporal/spatial embedding for video, (3) temporal/frequency embedding for audio, and (4) vision/audio patch embedding. As illustrated by the red and blue boxes in Fig. 2, the modality embeddings are two trainable vectors added to the input embeddings and used to indicate whether the input is from vision or audio input. In what follows, we explain the details of vision and audio embeddings.

**Vision Embeddings.** We adopt ViT [17]-style vision embedding, where each video frame of $224 \times 224$ pixels is divided into a list of $16 \times 16$-sized patches. Then, a liner projection layer is

applied to the normalized pixel values of each patch, resulting in a 768-dimensional patch embedding. For a video clip with N frame samples, the input tensor with shape $N \times 224 \times 224 \times 3$ (time $\times$ height $\times$ width $\times$ channel) will result in $N \times 14 \times 14$ embeddings. The temporal and spatial embeddings are different trainable vectors added to the time, height, and width axis of the $N \times 14 \times 14$ embeddings to incorporate the temporal and spatial information for each input patch. We treat image input as a single frame video so that our model can handle both image and video tasks without modification of the architecture [9]. Temporal embedding is only added for video inputs; we do not use temporal embedding for images.

**Audio Embeddings.** To obtain audio embeddings, we first convert the 1D waveform of the raw audio signal to 128-dimensional log Mel-spectrogram having a dimension of $T \times 128$ (time axis $\times$ frequency axis).[2] Then, we treat the audio spectrogram as an image, divide the spectrogram images into patches, and apply a liner projection layer on each patch to obtain a 768-dimensional patch embedding. This follows the audio embedding methods in recent work [19; 20; 7], where a similar modality-agnostic transformer is used to model spectrogram patches. We experiment with two different patch sizes: $16 \times 16$ (square patches similar to the vision modality) and $2 \times 128$ (the same area as the first one but covers the entire frequency domain with a shorter time range) and use trainable temporal and frequency embeddings to indicate the temporal and frequency information of patches.[3]

## 3.2 Multimodal Encoder-Decoder

The main architecture of TVLT is a transformer [80] consisting of a 12-layer encoder (hidden size 768), $E$, and an 8-layer decoder (hidden size 512), $D$. We follow He et al. [26] and use a shallow decoder that only serves for masked autoencoding objective (Sec. 4.2) and has much fewer computations than the encoder. After pretraining, we only use the encoder representation for finetuning on downstream tasks.

# 4 Pretraining Objectives

By virtue of our minimal and modality-agnostic design, TVLT is pretrained with two objectives: (1) vision-audio matching (Sec. 4.1) and (2) masked autoencoding (Sec. 4.2). For each training batch, we compute each objective through a separate forward pass and use the weighted sum of them for the final loss, where $\lambda^{\text{VAM}} = 1.0$ and $\lambda^{\text{MAE}} = 0.3$.

$$loss = \lambda^{\text{VAM}} loss^{\text{VAM}} + \lambda^{\text{MAE}} loss^{\text{MAE}} \tag{1}$$

## 4.1 Vision-Audio Matching

We use the vision-audio matching (VAM) objective to learn the global cross-modal representation, as illustrated in Fig. 2 (a). For each video input, we create a (positive) vision-audio pair $(x^{V+}, x^A)$. Then, we construct half of the vision-audio pairs inside a batch as mismatched (negative) pairs $(x^{V-}, x^A)$, by replacing video frames $x^{V+}$ with randomly sampled video frames $x^{V-}$ from the training dataset.

Following previous vision-and-language transformers [75; 10; 48; 32], a linear layer with sigmoid activation is used as the classification head applied to the encoder output of the first [CLS] token to obtain the matching probability $p$. Then we compute the binary cross-entropy loss as:

$$loss^{\text{VAM}} = -y \log p \tag{2}$$

where $y$ is 1 when the input vision-audio pair $(x^V, x^A)$ is matched and 0 otherwise.

## 4.2 Masked Autoencoding

In addition to the VAM objective to learn cross-modal representation, we also use the masked autoencoding (MAE) objective to improve unimodal representations in the vision-and-language

---

[2]We use `melspectrogram` method of `librosa` [50] with arguments: `sampling rate=44100`, `n_fft=2048, hop length=512, window='hann', pad_mode='constant', n_mels=128`.

[3]With 16x16 patch, a 20-second audio will have a spectrogram with shape $640 \times 128$ (time axis $\times$ frequency axis), resulting in $40 \times 8 = 320$ patches.

settings, by masking random patches of visual frames and the audio spectrogram, and reconstruct missing inputs as shown in Fig. 2 (b). Concretely, we randomly drop a portion of visual $x^V$ and audio embeddings $x^A$, then feed the remaining patch embeddings to the encoder $E$. We create inputs for the decoder $D$ by adding the dropped embeddings as trainable vectors [MASK] to the same location as the original input (gray boxes in Fig. 2 (b)). We also add the corresponding temporal, positional, and frequency embeddings to the decoder input. Note that the temporal, positional, and frequency embeddings of the encoder and decoder are separately parameterized. We calculate the mean squared error between the reconstructed and original video frames and spectrograms:

$$loss^{\text{MAE}} = \frac{1}{N_M^V} \sum_{i \in masked} ||x_i^V - \hat{x}_i^V||_2^2 + \frac{1}{N_M^A} \sum_{j \in masked} ||x_j^A - \hat{x}_j^A||_2^2 \qquad (3)$$

where $N_M^V$ and $N_M^A$ are the number of masked patches for vision and audio, respectively. We compute the loss only on masked patches, similar to BERT [15].

To save computation, we slice the audio and video parts of the encoder output and feed them separately to the decoder, rather than decoding the video frames and the audio spectrogram jointly. In Sec. 6.6, we show that separate decoding achieves better finetuning performance, as well as better efficiency than joint decoding.

### 4.3 Masking Strategy

**Vision Masking.** Following MAE [26], we randomly mask 75% of the visual patches, and the masking is applied for each video frame independently.

**Audio Masking.** Following MAE-AST [7], we randomly mask 75% of the spectrogram patches. To better capture speech-related audio representation, we emphasize audio masking on speech audios. We use Audiotok [3], an audio activity detection tool, to determine speech spans based on the detection of events in the energy of the audio signal. Then, we apply the masking only on those audio spans. We use a probability of 15%. We include the details of speech span detection in appendix.

## 5 Experimental Setup

To compare the audio-based and text-based language representations for vision-and-language tasks, we pretrain our TVLT and its text-based counterpart on video datasets. Then, we finetune the models on a set of downstream vision-and-language datasets for evaluation.

### 5.1 Text-based TVLT Counterpart

Our text-based TVLT counterpart has the same architecture as the vanilla TVLT with minor changes to accommodate text-based inputs. Firstly, we use sentence-piece [36] tokenizer and then map each token to trainable vectors to encode the raw text into embeddings, instead of converting the continuous input of frames or spectrograms into patch embeddings as in vanilla TVLT. Secondly, we follow the norm in mask language modeling [15] to use an affine layer as the decoder to recover masked words and set the mask ratio on text to be 15%, instead of using a transformer decoder to reconstruct 75% of the masked video and audio embeddings in vanilla TVLT.

### 5.2 Pretraining Datasets

**HowTo100M.** We used HowTo100M [52], a dataset containing 136M video clips of a total of 134,472 hours from 1.22M YouTube videos to pretrain our model. Our vanilla TVLT is pretrained directly using the frame and audio stream of the video clips. Our text-based TVLT is trained using the frame and caption stream of the video. The captions are automatically generated ASR provided in the dataset. We used 0.92M videos for pretraining, as some links to the videos were invalid to download.

**YTTemporal180M.** YTTemporal180M [86] includes 180M video segments from 6M YouTube videos that spans multiple domains, and topics, including instructional videos from HowTo100M [52], lifestyle vlogs of everyday events from the VLOG dataset [29], and YouTube's auto-suggested videos for popular topics like 'science' or 'home improvement'. Each video segment consists of 1) an image

frame extracted from the middle timestep of the segment, and 2) an ASR-based caption of L=32 BPE [18; 67] tokens. For each sample, we randomly sample a 15s video clip from the entire video to form a setting similar to HowTo100M dataset. Concretely, the original dataset provides 100 label files which are random split of the dataset. We sample 20% of YTTemporal180M (0.93M videos) so that the resulting subset consists of a similar number of videos to HowTo100M (0.92M videos), and call it YTT-S. In appendix, we show that pretraining TVLT on YTT-S can improve the downstream task performance of over pretraining on HowTo100M.

## 5.3 Downstream Tasks

We evaluate models on video-based and image-based vision-and-language tasks to compare the learned representation based on audio and text. For video-based tasks, we experiment with video retrieval [81; 90; 92] and multimodal sentiment analysis [84]. For image-based tasks, we experiment with image retrieval [83] and visual question answering [4; 21]. Although audio comes naturally with video, image-based tasks, such as visual question answering, do not include audio. Thus, we obtain audio queries for visual question answering via the text-to-speech (TTS) synthesis method (Sec. 5.4).

**Audio-to-Video Retrieval.**  Following AVLnet [63], we use MSR-VTT [81], Youcook2 [90], and CrossTask [92] for audio-to-video retrieval. We also follow the same data split in AVLnet [63] to finetune our models on their respective training set.

MSR-VTT is an open domain video dataset, consisting of 10,000 video clips from 20 categories such as music, movies or food. We follow AVLnet for the standard split, i.e., 6,783 training clips and 1000 test clips (where 32 videos do not have sound). We report the test split results.

Youcook2 is a video dataset on cooking tutorials that contains 2,000 long videos of 89 cooking recipes. Each recipe has on average 22 videos. It has 9,586 training clips and 3,350 validation clips. We report the validation split results.

CrossTask dataset contains instructional videos for 83 different tasks, divided into 18 primary tasks and 65 related tasks. Primary tasks are manually collected with temporal step human annotations and are the main focus of tasks such as cooking or repairing. Related tasks are automatically collected without any annotations and are tasks related to the primary tasks, such as masking latte (primary) vs. making machiato (related). The goal of related tasks is to assess whether they can improve primary tasks. It has 17,840 training clips and 2,819 validation clips. We report the validation split results. For all three tasks, we extract mp3 audio from videos with a sample rate of 44.1kHz. We also used the extracted audio or its corresponding ASR as retrieval queries for our experiment.

**Multimodal Sentiment / Emotion Analysis.**  We use CMU-MOSEI [84] for multimodal sentiment analysis. The dataset is made up of 23,454 movie review clips with more than 65.9 hours of YouTube video by 1000 speakers that cover 250 distinct topics. Each video clip also comes with a ground-truth transcription written by the author of the video. Following previous studies, we use the 15,288/4,830 train-test split and report the binary accuracy (A2) for sentiment analysis and weighted accuracy (WA) and F1 score on emotion classification over 6 emotion categories.

**Audio-to-Image Retrieval.**  We use Places-400k (The Places Audio Caption 400K Corpus) [25; 23; 24] for audio-to-image retrieval. The dataset contains approximately 1,000 hours of 400,000 spoken English captions for natural images drawn from the Places-205 [88] image dataset. The queries are conceptual descriptions of the image. The dataset also provides ASR of these audios. Places-205 is a large-scale scene dataset with 205 scene categories such as forest, bedroom, and coast, which contains 2,500,000 images in total.

**Visual Question Answering.**  We use VQAv1 [4] and VQAv2 [21] for visual question answering. VQAv1 contains 204,721 images from COCO [44] and 430,725 questions. VQAv2 is a newer version of VQAv1, with 265,016 images from COCO and 1,105,904 questions. For experiments with audio questions, we generate speech audio from textual questions using TTS (Sec. 5.4) and report test-dev results for both tasks.

**Finetuning on Downstream tasks.**  For each of the downstream tasks, we add a task-specific head (two-layer MLP) on top of the encoder representation. For retrieval tasks, we use an MLP to map

Table 1: Comparison of TVLT and its text-based counterpart on audio-to-video retrieval and video-based multimodal sentiment analysis benchmarks; *HT100M*=HowTo100M, *YTT-S*=YTTemporal180M subset.

| Method | Input Mod. | | | Pretrain Datasets | Audio-to-Video Retrieval (R@1) ↑ | | | Sentiment (A2) ↑ | Latency ↓ |
|--------|---|---|---|---|---|---|---|---|---|
| | V | T | A | | MSR-VTT | Youcook2 | CrossTask | CMU-MOSEI | (ms) |
| TVLT | ✓ | ✓ | | - | 3.1 | 5.0 | 2.2 | 68.1 | 2916 |
| TVLT | ✓ | | ✓ | - | 4.3 | 4.7 | 2.7 | 65.7 | 103 |
| TVLT | ✓ | ✓ | | HT100M | 17.1 | 24.9 | 11.1 | 76.5 | 2916 |
| TVLT | ✓ | | ✓ | HT100M | 22.6 | 31.8 | 14.9 | 75.3 | 103 |
| TVLT | ✓ | ✓ | | YTT-S | 19.3 | 26.3 | 12.2 | 76.6 | 2916 |
| TVLT | ✓ | | ✓ | YTT-S | **23.8** | **32.8** | **15.3** | **76.8** | 103 |

encoder representation of [CLS] to matching scores $\in [0, 1]$, which correspond to match vs. mismatch pairs, and train the model jointly with binary cross-entropy loss. For visual question answering tasks, we use an MLP to map the encoder representation of [CLS] to the answer probabilities with 3129 answer candidates, and train the model jointly with binary cross-entropy loss in a multi-label classification setup. For multimodal sentiment analysis tasks, we use an MLP to map the encoder representation of [CLS] token to the entiment scores, and train the model jointly with L2 regression loss.

## 5.4 Other Details

**Automatic Speech Recognition (ASR).** For the text-based model mentioned above, we obtain text from audio with different automatic speech recognition (ASR) models. We use the `asr-crdnn-rnnlm-librispeech` ASR model from the Speechbrain package [62]. The model is based on RNN language model and CRDNN encoder-CTC/Attention decoder architecture and is trained on LibriSpeech [57]. We also experiment with the Google Cloud Speech-to-Text API which uses Conformer [22] as the backend model.[4]

**Text-to-Speech (TTS).** We use `WaveNet` [79] Google Cloud Text-to-Speech API[5] to generate audio input for the questions in VQAv2. Since VQAv2 questions are written in English, we use a `en-US-neutral` speaker. We follow the default pitch and speech configuration. We use the `mp3` audio format with a sample rate of 44.1kHz to match the audio configuration used in the pretraining.

**Pretraining.** We train TVLT and the text-based TVLT counterpart for 200k steps using Adam optimizer [33] with a learning rate of 1e-5, batch size 4096, and a decay rate of 0.001 with a cosine schedule [47]. We initialize the weights of both models with the masked autoencoder transformer in He et al. [26] that is pretrained on ImageNet [14]. For the pretraining objectives in Eq. (1), we use $\lambda^{\mathrm{VAM}} = 1.0$ and $\lambda^{\mathrm{MAE}} = 0.3$. For each video clip, we uniformly sample 8 frames. Pretraining takes 2 weeks with 4 NVIDIA RTX A6000 GPUs (each 49GB memory).

**Finetuning on Downstream Tasks.** We use a learning rate of 1e-5, batch size 256, and a decay rate of 0.001 with a cosine schedule for all tasks. For each video clip, we uniformly sample 8 frames. We use 2 NVIDIA RTX A6000 GPUs.

## 6 Results and Analysis

### 6.1 Comparison to Text-based Counterpart

Table 1 shows that TVLT outperforms the text-based counterpart in audio-to-video retrieval tasks when pretrained on either HowTo100M or YTT-S. On CMU-MOSEI sentiment analysis, TVLT also outperforms its text variant when pretrained on YTT-S. In Table 2, although TVLT slightly underperforms the text-based counterpart on audio-to-image retrieval and visual question answering, TVLT can still achieve decently comparable results and remain competitive while being 27x faster during inference due to the removal of ASR from the processing pipeline. More details on

---

[4] https://cloud.google.com/speech-to-text
[5] https://cloud.google.com/text-to-speech/docs/wavenet

Table 2: Comparison of TVLT and its text-based counterpart on audio-to-image retrieval and visual question answering benchmarks.

| Method | Input Mod. | | | Pretrain Datasets | Audio-to-Image Retrieval | Visual QA (Acc.) ↑ | Latency ↓ |
| | V | T | A | | Places-400k (R@1 / R@5 / R@10) ↑ | VQAv2 | (ms) |
|---|---|---|---|---|---|---|---|
| TVLT | ✓ | ✓ | | - | 13.0 / 35.9 / 49.7 | 47.0 | 2010 |
| TVLT | ✓ | | ✓ | - | 12.7 / 33.3 / 48.0 | 46.7 | 52 |
| TVLT | ✓ | ✓ | | HT100M | 50.4 / 78.2 / 87.0 | 62.1 | 2010 |
| TVLT | ✓ | | ✓ | HT100M | 48.7 / 77.9 / 86.0 | 60.8 | 52 |
| TVLT | ✓ | ✓ | | YTT-S | **54.3 / 78.9 / 88.8** | **63.2** | 2010 |
| TVLT | ✓ | | ✓ | YTT-S | 49.0 / 78.2 / 86.8 | 61.0 | 52 |

efficiency analysis are given in Sec. 6.2. The results provide evidence of the possibility of learning a more compact and efficient vision-and-language representation from raw visual and audio signals compared to the prevailing VL learning paradigms with explicit text-based modules in the pipeline.

## 6.2 Efficiency Comparison

To test inference latency, we sample 100 videos in CMU-MOSEI. As the average video length in the CMU-MOSEI dataset is 12 seconds, we measure the latency with two sets of input video lengths: 10 and 20 seconds. For 10s and 20s videos, we also use 4 and 8 video frames, respectively. Then we calculate the processing time of Fast Fourier Transform (FFT),

Table 3: Latency of FFT, ASR and VL Models.

| Model | # Param | Video Input | Latency (ms) ↓ | | | |
| | | Length / # Frames | FFT | ASR | VL | Total |
|---|---|---|---|---|---|---|
| ASR-SpBr | 195M | 10s / 4 | - | 2110 | - | - |
| | | 20s / 8 | - | 2890 | - | - |
| TVLT | 88M | 10s / 4 | 40 | - | 40 | 80 |
| | | 20s / 8 | 60 | - | 43 | 103 |
| TVLT + text | 88M + 195M | 10s / 4 | - | 2110 | 25 | 2135 |
| | 88M + 195M | 20s / 8 | - | 2890 | 26 | 2916 |
| AVLnet | 158M | 10s / 4 | 40 | - | 208 | 248 |
| AVLnet + text | 158M + 195M | 10s / 4 | - | 2110 | 206 | 2316 |

SpeechBrain (ASR-SpBr) [62], TVLT, text-based TVLT, and AVLNet on the sampled inputs. Speech-Brain is the default ASR module that we used in our text-based counterpart pipeline (see Sec. 5.4).

As shown in Table 3, we find that ASR dominates the inference time for text-based models. Although ASR helps reduce the input length in transformers (as indicated by the VL module latency decrease), TVLT is more than 27x and 28x faster than text-based TVLT for inference with video input lengths of 10s and 20s, respectively, with only 1/3 of the parameters. The comparison is also shown in Fig. 1. In the bottom rows, we also show the inference latency of AVLnet and its text variant, where TVLT is 3x faster than AVLnet which contains audio-specific convolution modules.

## 6.3 Text Query vs. Speech Query for Language-based Video Retrieval

For text-to-video retrieval tasks, text captions are commonly used for queries [81]. In Sec. 6.1, we show the experiment of audio-to-video retrieval tasks following AVLnet [63], where the audio queries are the sounds of the original videos. Since video sounds and text captions have different information, the audio-to-video retrieval results are not directly comparable to the results in other text-to-video retrieval papers. For a

Table 4: Text vs. Speech Query for Video Retrieval.

| Method | Pretrain Datasets | Query | Video Retrieval (R@1) ↑ |
| | | | MSR-VTT |
|---|---|---|---|
| TVLT | HT100M | Caption | 22.0 |
| TVLT | HT100M | Speech Audio (TTS) | 20.1 |
| HERO [42] | HT100M | Caption | 16.8 |
| DeCEMBERT [76] | HT100M, TVQA | Caption | 17.5 |
| ClipBERT [39] | COCO, VG | Caption | 22.0 |
| AVLnet [63] | HT100M | Caption | 22.5 |

better comparison, we experiment with video retrieval based on two language queries: 1) text captions and 2) speech audio obtained by TTS (see Sec. 5.4) from text captions. Table 4 shows MSR-VTT video retrieval results of TVLT with text/audio queries and recent text-to-video retrieval models pre-trained with a similar scale of data.[6] Although TVLT with audio query slightly underperforms its text query counterpart due to TTS errors, it still outperforms other text-to-video retrieval models (HERO [42] and DeCEMBERT [76]), showing promising possibilities of speech-based video retrieval.

---

[6]We exclude the models pretrained on large-scale image captions such as Conceptual Captions [68] that has written annotation, or visual encoder pretrained on a large-scale dataset beyond the scale of ImageNet [14], such as CLIP [59], as they are not directly comparable to our models.

Table 5: TVLT on CMU-MOSEI emotion analysis test set; *WA*=weighted accuracy, *F1*=weighted f1.

| Method | Input Mod. | | | Happy | | Sad | | Angry | | Fear | | Disgust | | Surprise | |
|---|---|---|---|---|---|---|---|---|---|---|---|---|---|---|---|
| | V | T | A | WA | F1 | WA | F1 | WA | F1 | WA | F1 | WA | F1 | WA | F1 |
| TVLT | ✓ | ✓ | | 64.7 | 63.9 | 70.2 | 66.0 | 68.9 | 71.8 | 66.2 | 84.4 | **70.7** | **82.9** | 58.4 | 86.2 |
| TVLT | ✓ | | ✓ | **65.1** | **64.1** | **72.2** | **70.0** | **69.9** | **72.1** | **68.1** | **88.0** | 68.8 | 79.6 | **62.1** | **87.4** |

## 6.4 Emotion Analysis

Since TVLT takes raw visual and audio input instead of relying solely on text as in text-based TVLT, we further investigate what type of information TVLT can learn beyond speech on CMU-MOSEI emotion classification task. As shown in Table 5, TVLT outperforms the text-based counterpart in most emotion categories, except for 'Disgust'. We conjecture that TVLT is capable of capturing speech-related acoustic information, such as tone and loudness, which is helpful in recognizing these emotions, while this ability is absent from text-based ASR-dependent models.

Table 6: Finetuning performance on audio-to-video retrieval and multimodal sentiment analysis benchmarks. For a fair comparison, we gray out the models that use ground-truth text transcription as additional input for CMU-MOSEI.

| Method | Input Mod. | | | Pretrain Datasets | Audio-to-Video Retrieval (R@1) ↑ | | | Sentiment (A2) ↑ |
|---|---|---|---|---|---|---|---|---|
| | V | T | A | | MSR-VTT | Youcook2 | CrossTask | CMU-MOSEI |
| Multilogue-Net [69] | ✓ | | ✓ | - | - | - | - | 75.2 |
| AVLnet [63] | ✓ | | ✓ | HT100M | 20.1 | 30.7 | 13.8 | - |
| TVLT (Ours) | ✓ | | ✓ | HT100M | 22.6 | 31.8 | 14.9 | 75.3 |
| TVLT (Ours) | ✓ | | ✓ | YTT-S | **23.8** | **32.8** | **15.3** | **76.8** |

Table 7: Finetuning performance on audio-to-image retrieval and visual question answering (Visual QA). For Visual QA, we create spoken questions from text via TTS (Sec. 5.4). [†]CSC (Conceptual Spoken Caption) is 3.3M image-speech pairs, where speech is obtained via TTS API from Conceptual Captions. The CSC dataset is not publicly available.

| Method | Input Mod. | | | Pretrain Datasets | Audio-to-Image Retrieval | Visual QA (Acc.) ↑ |
|---|---|---|---|---|---|---|
| | V | T | A | | Places-400k (R@1 / R@5 / R@10) ↑ | VQAv1 / VQAv2 |
| TextMod [87] | ✓ | ✓ | | - | - | 56.7 / - |
| SpeechMod [87] | ✓ | | ✓ | - | - | 47.0 / - |
| AVLnet [63] | ✓ | | ✓ | HT100M | 44.8 / 76.9 / 86.4 | - |
| MILAN [64] | ✓ | | ✓ | CSC[†] | **53.4 / 79.1** / 86.3 | - |
| TVLT (Ours) | ✓ | | ✓ | HT100M | 48.7 / 77.9 / 86.0 | 58.6 / 60.8 |
| TVLT (Ours) | ✓ | | ✓ | YTT-S | 49.0 / 78.2 / **86.8** | **58.9 / 61.0** |

## 6.5 Comparison to State-of-the-art Textless Models

We compare our TVLT with recent models that also take raw visual and audio signals as input but involve audio-specific designs in their networks. As shown in Table 6, TVLT outperforms AVLnet [63] on three audio-to-video retrieval (MSR-VTT, Youcook2, CrossTask) tasks and outperform Multilogue-Net [69] on multimodal sentiment analysis (CMU-MOSEI) task with a simple modality-agnostic design. Similarly, Table 7 shows that TVLT achieves competitive results with AVLnet [63] and MILAN [64] on audio-to-image retrieval (Places-400k). Note that MILAN[7] is pretrained on Conceptual Spoken Caption [30] which contains 3.3M well-aligned image-speech pairs taken from Conceptual Captions [68] with TTS generated speech, whereas our TVLT is able to elicit effective representation from video inputs where vision-and-language clues are only weakly aligned. TVLT also outperforms both variants of the VQA models (TextMod, SpeechMod) in Zhang et al. [87] on VQAv1.

## 6.6 Ablation Studies

In the following, we show the results of the ablation study on TVLT training details: the audio masking strategy, the encoder/decoder architectures, and the pretraining objectives.

---

[7]The dataset is also not publicly available.

**Audio Masking Strategy.** In Table 8, we show the result of finetuning performance with different audio masking configurations, described in Sec. 4.3. For patch sizes, masking audio patches on detected speech spans improves performance across the board. However, we did not observe strict superiority between the two patch sizes; $2 \times 128$ achieves higher scores on MSR-VTT, while $16 \times 16$ achieves higher scores on VQAv2. For our default pretrain-

Table 8: Audio masking configurations.

| Patch Size | Masking on speech | MSR-VTT (R@1) | VQAv2 (Acc.) |
|---|---|---|---|
| $16 \times 16$ | | 21.7 | 57.8 |
| $16 \times 16$ | ✓ | **22.3** | 58.6 |
| $2 \times 128$ | | 21.0 | 58.8 |
| $2 \times 128$ | ✓ | 21.2 | **59.2** |

ing configuration, we use the $16 \times 16$ patch size and use speech span detection, since the $16 \times 16$ sized patch is also used in visual embedding (thus modality-agnostic) and speech span detection improves performance with minimal additional computation (see appendix).

**Encoder Architecture.** As described in Section 3.2, we use the joint encoder in TVLT. We compare this to modality-specific encoders for vision and audio. Table 9 below compares the separate encoders with the joint encoder for two tasks: VQAv2 and MSR-VTT. To tackle VQAv2 with separate encoders, we learned a two-layer self-attention fusion layer over the concatenation of hidden states of the vision and audio

Table 9: Encoder variants.

| Encoder | MSR-VTT (R@1) | VQAv2 (Acc.) |
|---|---|---|
| Separate | 9.6 | 53.1 |
| Joint | **10.2** | **54.6** |

encoder. Our joint encoder architecture achieves better accuracy on both tasks than a separate encoder architecture. The results show that although vision and audio spectrogram are two different modalities, the single joint encoder could learn useful cross-modal representation for VL tasks without needing modality-specific encoders.

**Decoder Architecture.** As described in Sec. 4.2, we use separate decoders (with shared weights) for the vision and audio MAE pretraining objectives. We compare this separate decoding with joint decoder, where we feed the concatenated encoder outputs to the decoder and jointly reconstruct the video frames and spectrogram. Table 10 shows that pretraining with separate decoder outperforms joint decoder on finetuning performance, while being more efficient as well.

Table 10: Decoder variants.

| Decoder | MSR-VTT (R@1) | VQAv2 (Acc.) |
|---|---|---|
| Separate | **22.3** | **58.6** |
| Joint | 22.0 | 58.1 |

**Pretraining Objectives.** We measure the impact of each pretraining objective described in Sec. 4. Table 11 shows that each of the pretraining objectives (MAE and VAM) improves finetuning performance over random weight initialization. The combination of VAM and MAE further improves the finetuning performance, and we use this configuration as default for TVLT pretraining.

Table 11: Pretraining objectives.

| Objectives | MSR-VTT (R@1) | VQAv2 (Acc.) |
|---|---|---|
| Random init | 4.3 | 46.7 |
| VAM | 21.0 | 56.2 |
| MAE | 18.6 | 54.1 |
| VAM + MAE | **22.3** | **58.6** |

## 7 Conclusion

In this work, we present TVLT, a simple end-to-end vision-and-language transformer that can accept low-level visual and audio signals for vision-and-language representation learning. Our TVLT achieves competitive performance with other state-of-the-art audio-based vision-and-language models on visual question answering, image retrieval, video retrieval, and multimodal sentiment analysis. We also show that by eliminating the need for expensive ASR in the model pipeline, TVLT can be 28x faster than its text-based counterpart while achieving comparable performance. We comprehensively analyze the efficiency of our model and show ablation studies over different training variants. We hope that our research will inspire further exploration of simple and efficient vision-and-language frameworks with low-level signals.

## Acknowledgments

We thank the reviewers for their helpful comments. This work was supported by ARO Award W911NF2110220, DARPA KAIROS Grant FA8750-19-2-1004, ONR Grant N000141812871, and NSF-AI Engage Institute DRL-211263. The views, opinions, and/or findings contained in this article are those of the authors and not of the funding agency.

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
