# Supplementary Materials for
# TVLT: Textless Vision-Language Transformer

**Zineng Tang***   **Jaemin Cho***   **Yixin Nie***   **Mohit Bansal**
UNC Chapel Hill
{terran, jmincho, yixin1, mbansal}@cs.unc.edu

In this appendix, we include the pretraining dataset combination experiment (Appendix A), ASR quality experiment (Appendix B), implementation details (Appendix C), finetuning on unimodal ASR task (Appendix D), visualization of MAE reconstruction (Appendix E), limitations and potential negative impacts (Appendix F), and licenses (Appendix G).

## A Combination of Pretraining Datasets

Table 1 and Table 2 in the main paper show that TVLT either pretraining on HowTo100M [2] or YTT-S [5] can outperform random initialization across the board. Among the two pretraining datasets, models pretrained on YTT-S achieve higher performance than models pretrained on HowTo100M. The relative improvement is consistent with the findings of Zellers et al. [5], and we suspect that coverage of a wider range of video topics improves overall performance. We also experiment with pretraining TVLT with the combination of HowTo100M and YTT-S. The total size of the pretraining dataset size is 1.85M = (0.92M + 0.93M) videos, and we pretrain the model for 200k steps. As shown in Table 1, pretraining on the combination of both datasets achieves better finetuning performance than single-dataset pretraining on both MSR-VTT audio-to-video retrieval and VQAv2. The results indicate that TVLT can take advantage of the domain diversity of YTT-S and that pretraining with data from a diverse range of domains can result in a more adaptable representation.

Table 1: Finetuning performance of TVLT pretrained on different datasets.

| Method | Input Mod. | | | Pretrain Datasets | Audio-to-Video Retrieval (R@1) ↑ | Visual QA (Acc.) ↑ |
|--------|---|---|---|-------------------|----------------------------------|--------------------|
| | V | T | A | | MSR-VTT | VQAv2 |
| TVLT | ✓ | | ✓ | HowTo100M | 22.6 | 60.8 |
| TVLT | ✓ | | ✓ | YTT-S | 23.8 | 61.0 |
| TVLT | ✓ | | ✓ | HowTo100M+YTT-S | **25.0** | **61.4** |

## B Impact of ASR quality

Table 2 shows the results of TVLT on CMU-MOSEI sentiment analysis with the following different inputs: audio, ASR-based text, and ground-truth text transcriptions. ASR-Google and ASR-SpeechBrain refer to Google Cloud API and SpeechBrain, respectively (see main paper Sec. 5.4). Although TVLT pretrained on HowTo100M underperform the text variant with SpeechBrain ASR input, TVLT pretrained on YTT-S (76.8) achieves comparable results to

Table 2: TVLT with audio/text on CMU-MOSEI.

| Language Input | CMU-MOSEI (A2) ↑ | |
|----------------|---------|--------|
| | HT100M | YTT-S |
| Audio | 75.3 | **76.8** |
| Text (ASR-SpeechBrain) | **76.5** | 76.6 |
| Text (ASR-Google) | 77.1 | 77.8 |
| Text (GT Transcripts) | 78.9 | 79.1 |

---

*equal contribution

36th Conference on Neural Information Processing Systems (NeurIPS 2022).

those of the text variant with SpeechBrain ASR (76.6), which sheds light on the effectiveness of TVLT. Although there is still a gap between TVLT and text-based TVLT with higher quality ASR or ground truth transcript input, we expect that TVLT can be further improved with larger-scale pretraining (e.g., full YTTemporal180M dataset) on raw video signals.

To better understand the impact of ASR on downstream tasks, we show two examples of the CMU-MOSEI sentiment analysis task in Table 3. For example (a), ASR-Google Cloud provides more accurate transcription than ASR-SpeechBrain, resulting in more accurate sentiment estimation (ASR-SpeechBrain: -1.0 vs. ASR-Google: 0.0; label: 0.0). For example (b), ASR-Google Cloud and ASR-SpeechBrain provide similar transcription quality, resulting in the same sentiment estimation (ASR-SpeechBrain: 2.0 vs ASR-Google: 2.0; label: 1.0).

Table 3: Comparison of different ASR models on CMU-MOSEI sentiment analysis task. Sentiment label has range $[-3, 3]$, where -3 and 3 corresponds to negative and positive, respectively. We use TVLT pretrained on HowTo100M.

| | GT Transcripts | ASR-SpeechBrain | ASR-Google Cloud | Pred (GT Transcripts) | Pred (ASR-SpBr) | Pred (ASR-GC) | Label |
|---|---|---|---|---|---|---|---|
| (a) | This is a new movie (uhh) in which a character is confined to his house, he is under house arrest, and his mother takes away his Xboxes and TV as sort of a little bit of additional punishment | communicate of additional punishment thoroughly | Serbia this is a new movie and which a character is confined to his house. He is under house arrest and his mother takes away his Xboxes and TVs is sort of a little bit of additional punishment. | 0.0 | -1.0 | 0.0 | 0.0 |
| (b) | The club that I'm part of that organize that has about currently 40 some students and then last year we had 260-something come out to the dance. | well the club that i'm part of that organizes it has about currently forty some students and then flashed here we had two hundred and sixteen something come out to the dance | The club that I'm part of that organizes it has about currently 40 some students. And then last year we had 260 something come out to the dance | 1.0 | 2.0 | 2.0 | 1.0 |

## C  Implementation Details

### C.1  Speech Span Detection

For the speech span detection mentioned in the main paper Sec. 4.3, we use the Audiotok [1] word-level speech event detector. We use the configurations as follows: (1) We set a single speech event to have a duration within [0.3s, 1.2s], so that an event is likely to cover a single word. (2) We set $\texttt{max\_silence} = 0.05s$. $\texttt{max\_silence}$ refers to the maximum silence gap between two speech spans. If the silence gap is too large, it is usually a stop between two words. Therefore, setting a low value ensures that we do not detect two words as a single word. (3) We use an energy threshold of 70, which is higher than the default value of 55, to avoid false positives of detecting noise. This is because real-world audio contains natural sounds and noises that usually come with a high level of audio signal energy. In the speech spans detected on HowTo100M, each word has an average length of 15 in our audio spectrogram (Sec. 3.1). As this is similar to the size of a single audio patch (16x16), masking an audio patch usually covers a word in speech.

Table 4: Audio Pipeline Latency.

| Audio Length | CPU Latency (ms) ↓ | | | GPU Latency (ms) ↓ | |
|---|---|---|---|---|---|
| | Data Loading | Fast Fourier Transform | Speech Span Detection | ASR | VL Model |
| 10s | 60 | 40 | 130 | 2110 | 40 |
| 20s | 110 | 60 | 170 | 2890 | 43 |

## C.2 Audio Pipeline latency

In Table 4, we show the detailed latency for each audio processing pipeline for two different audio length settings: 10s and 20s. In both settings, ASR takes significantly longer processing time than all other modules and becomes the bottleneck of the entire vision-and-language pipeline.

## D Finetuning on Unimodal ASR Task

To explore whether the cross-modal representation of TVLT is useful for unimodal tasks, we experiment using TVLT as an audio encoder for an ASR model. Specifically, we construct a 4-layer transformer language model that attends to TVLT encoder outputs via cross-attentions and jointly train the encoder and decoder. We experiment with two settings: where the TVLT encoder is randomly initialized or initialized with V+A pretraining. We train the models on LibriSpeech [3], a widely used ASR corpus with 960 hours of English audiobooks, and evaluate them on its two dev sets, dev-clean and dev-other. As shown in Table 5, our ASR model with V+A pretrained TVLT encoder outperforms the No-pretrain baseline by 0.8 (dev-clean) and 1.3 (dev-other) in Word Error Rate (WER), respectively. The results show that the cross-modal representation learned by TVLT could also be helpful for ASR, a unimodal task.

Table 5: Finetuning on ASR.

| Encoder PT | WER (%) ↓ | |
| --- | --- | --- |
| | dev-clean | dev-other |
| No-pretrain | 3.1 | 6.0 |
| V+A pretrain | **2.3** | **4.7** |

## E MAE Reconstruction Visualization

In Figure 1 and Figure 2, we show the reconstruction results with the MAE head, described in the main paper Sec. 4.2. In each figure, the left column shows the masked input, the middle column shows the reconstruction, and the right column shows the target. We use masking ratio 0.75, image size $224 \times 224$, and audio spectrogram size $176 \times 128$ (time $\times$ frequency) for this visualization.

## F Limitations

**Green AI.**  A key barrier to the adoption of Green AI [4] has been the incentive to use massive computational power for pretraining. As shown in our main paper, TVLT is also subject to pretraining in order to achieve decent performance on visual linguistic tasks. While TVLT is substantially faster than vision-and-language models with explicit text-based modules that can help reduce pretraining computation, there is still scope for future work on energy-efficient training to alleviate the heavy reliance on large-scale pretraining.

**English-only Datasets.**  We perform transfer learning with TVLT pretrained with HowTo100M and YTTemporal180M datasets. Both datasets mostly contain content in English, since HowTo100M [2] videos are obtained from English queries, and the authors of YTTemporal180M [5] filtered out videos with non-English ASR results. Therefore, our models pretrained with the two datasets might not have a good performance on non-English tasks without additional pretraining.

Note that the TVLT framework is a language-agnostic method, so one can adapt our model to a non-English dataset without any architectural change. Furthermore, our architecture eliminates the need for external ASR modules, which reduces the computation of the typical vision-and-language pipeline. To reduce environmental damage, we will publicly release our code and pretrained checkpoint.

## G License

We will publicly release our code and models. We use standard licenses from the community and provide the following links to the licenses for the datasets, codes, and models that we used in the project. For more details, see the individual link.

**HowTo100M:** Apache

**YTTemporal180M:** MIT

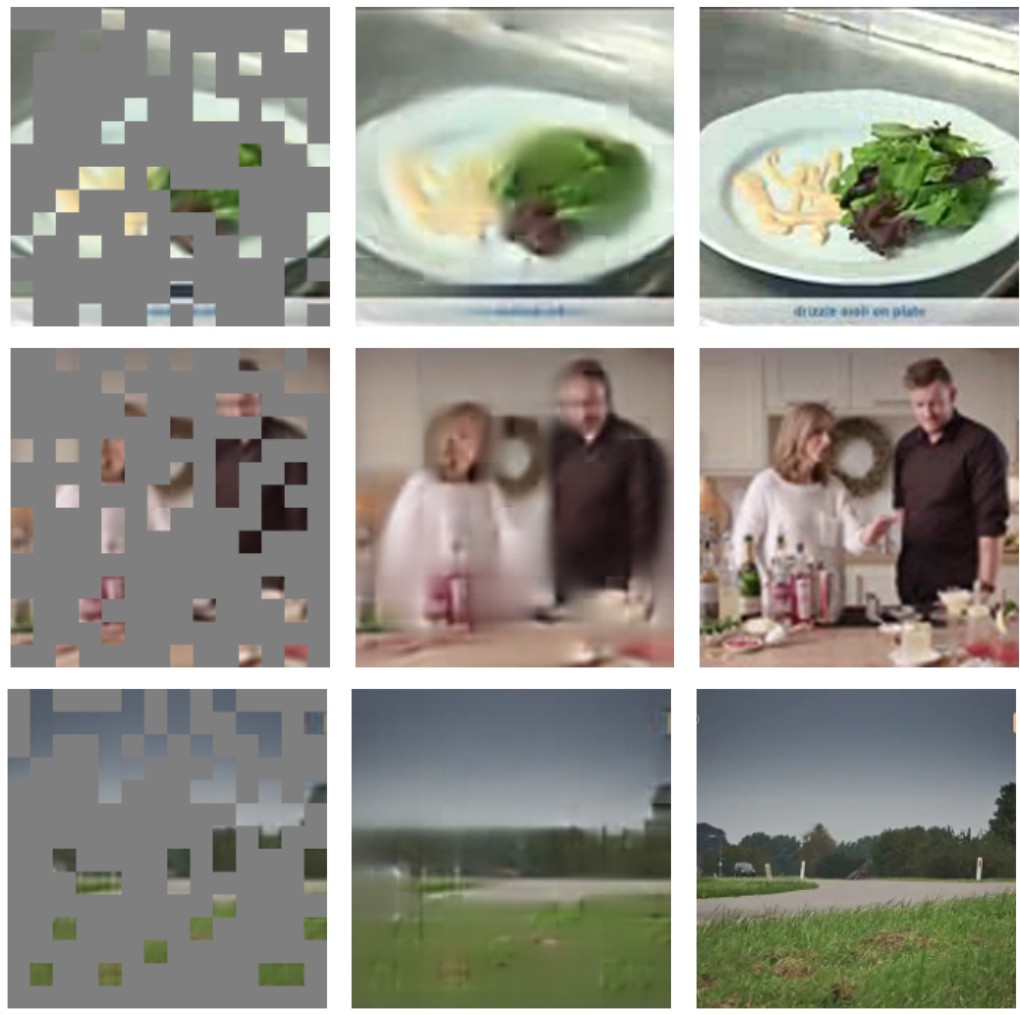

Figure 1: Visualization on video frames reconstruction (single frame): masked frames (left), reconstruction (middle), and original frames (right).

**PyTorch:** BSD-style

**Huggingface Transformers:** Apache

**Torchvision:** BSD 3-Clause

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

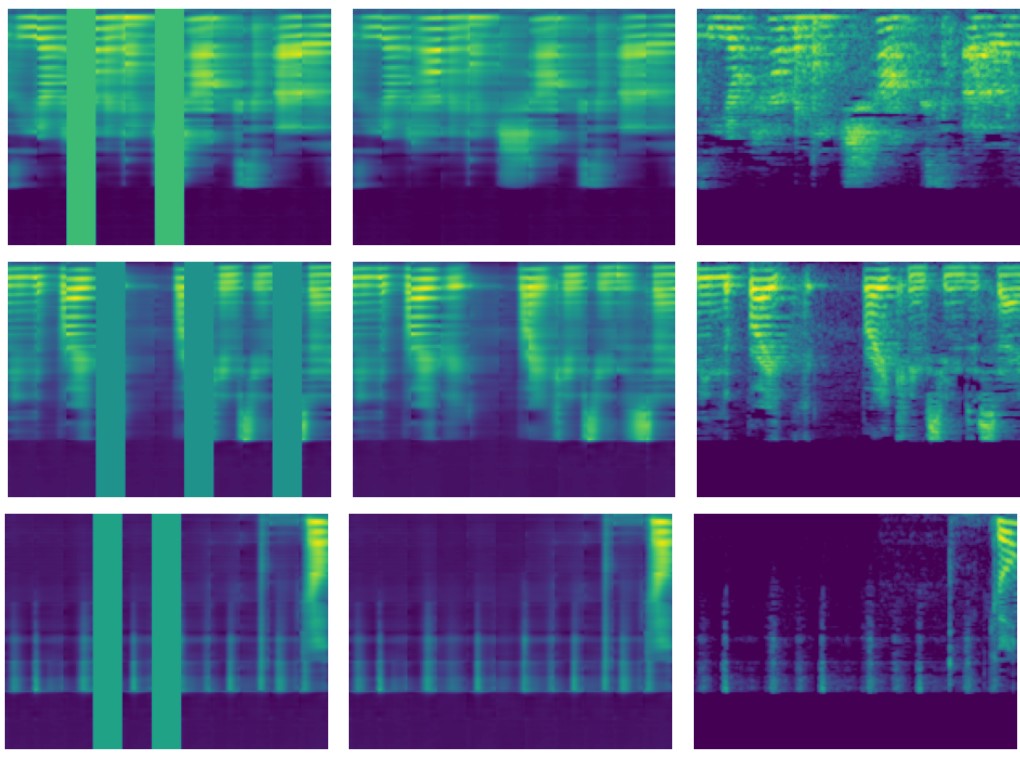

Figure 2: Visualization on video frames reconstruction: masked audio spectrogram (left), reconstruction (middle), and original audio spectrogram (right).