# OpenReview forum: "TVLT: Textless Vision-Language Transformer"
_NeurIPS.cc/2022/Conference — NeurIPS 2022 Accept_

### Official Review · Reviewer_JbWb · 2022-07-06

**Rating:** 8
**Confidence:** 4
**Soundness:** 3 good
**Presentation:** 4 excellent
**Contribution:** 3 good

**Summary:**

This paper presents an audio-visual multi-modal representation learning model called TVLT, and it shows the effectiveness of the learned representations on the downstream tasks that require the understanding of visual and linguistic information. Unlike the previous vision and language representation learning methods that take text input to help elicit the linguistic information, it takes raw audio input instead of the text input but still achieves comparable results with much more improved inference efficiency. In experiments, TVLT shows comparable performance to the text-based counterpart model on audio-to-video retrieval and video-based multimodal sentiment analysis. Although TLVT is slightly worse than the text-based counterpart model, it is still able to achieve comparable results while being 50x faster during inference. In addition, this paper conducts extensive experiments such as a comparison to the SOTA textless models and an ablation study on masking strategy and various model architectures.

**Questions:**

* Is it possible to use each of the representations for uni-modal subtasks (e.g. Can audio representations be used for ASR training)?
* What if the text-based model uses all the modalities; text, image, and audio? This is because, as it is mentioned in the paper, the audio modality contains more information that is absent in the text.

(I totally understand that it is almost impossible to conduct every experiment, but it is written somewhat as part of curiosity. So don't take these questions too negatively :D)

**Limitations:**

This paper adequately addressed the limitations and did considerations about the negative societal impact.

**Strengths And Weaknesses:**

### Strengths

- The paper is well-written and the flow of the writing is natural and neat so it was easy to read.
- The proposed method is original and it presents an interesting research direction for future works.
- I believe this work would give a significant impact on this field and its simple architecture is likely to be used by other people.
- Especially, the minimalist design of TVLT allows to scale up the model so it is expected to give another possibility of large-scale multi-modal model as GPT-3 did in NLP.

### Suggestions

- As far as I know, the tables should be centered in the papers submitted to NeurIPS so I think it should be fixed.
- There are several suggestions of the experiments that can be conducted in the future work. I listed them in the questions section below.
- In line 128, I think it might be a typo, $(x^{V+} → x^{V-})$
- It can give more insights if there are samples of the masked data instances.

---

> ### Author Response · Authors · 2022-08-02
> **Response to Reviewer JbWb**
>
> Thank you for appreciating the strengths of our paper and providing useful feedback.
>
> **(1) Is it possible to use each of the representations for uni-modal subtasks (e.g. Can audio representations be used for ASR training)?**:
>
> To explore whether the cross-modal representation of TVLT is useful for uni-modal tasks, we experiment with using TVLT as an audio encoder for an ASR model. Specifically, we construct a 4-layer transformer language model that attends to TVLT encoder outputs via cross-attentions and jointly train the encoder and decoder. We experiment with two settings: where the TVLT encoder is randomly initialized or initialized with V+A pretraining. We train the models on LibriSpeech, a widely used ASR corpus with 960 hours of English audiobook, and evaluate them on its two dev sets, dev-clean and dev-other.
>
> As shown in the table below, our ASR model with V+A pretrained TVLT encoder outperforms the No-pretrain baseline by 0.8 (dev-clean) and 1.3 (dev-other) in Word Error Rate (%, lower is better), respectively. The results show that the cross-modal representation learned by TVLT could also be helpful for ASR, a uni-modal task. We will add this in the extra page of the final version, thanks for the suggestion.
>
> | | TVLT encoder pretraining | dev-clean | dev-other |
> | --- | --- | --- | --- |
> | TVLT encoder + 4-layer transformer LM  | No-pretrain | 3.1 | 6.0 |
> | TVLT encoder + 4-layer transformer LM | V+A pretrain | **2.3**  | **4.7** |
>
> **(2) What if the text-based model uses all the modalities; text, image, and audio?**:
>
> We appreciate the reviewer's curiosity regarding models taking in all modalities. There has been a line of works [1, 2, 3] that combines all three input modalities: vision, text, and audio, achieving better results than models with two inputs: vision and text. Our work differs from them (as you mentioned), since we focus on demonstrating the possibility of using a minimalist design, where homogeneous transformer blocks process both raw visual and audio input from videos (Sec. 3), and presenting a novel textless vision-and-language (VL) representation learning method based on the design (Sec. 4). To answer your question and provide more empirical results in the community, we also experiment with CMU-MOSEI sentiment analysis by using all modalities.
>
>
> | | Finetuning on V+T | Finetuning on V+T+A |
> | --- | :---: | :---: |
> | No-pretrain |  76.1  |    76.9 |
> | V+A pretrain  |  80.2 |    81.9  |
> | V+T pretrain |  81.1  |    82.4  |
>
> The table above compares the TVLT on CMU-MOSEI sentiment analysis task in A2 (binary accuracy) metric with different pretraining/finetuning modality combinations. As expected, in all three pretraining setups (No-pretrain / V+A pretrain / V+T pretrain), using all three modalities (V+T+A) achieves a slightly better (within 1.5%) A2 than using only vision and text (V+T).
>
> - [1] Zellers et al., MERLOT Reserve: Neural Script Knowledge through Vision and Language and Sound, CVPR 2022
> - [2] Shenoy and Sardana, Multilogue-Net: A Context-Aware RNN for Multi-modal Emotion Detection and Sentiment Analysis in Conversation, ACL Workshop 2020
> - [3] Tsai et al., Multimodal Transformer for Unaligned Multimodal Language Sequences, ACL 2019
>
>
> **(3) Masked data instance**:
>
> We provide samples of masked video frames/audio spectrograms and their reconstruction in appendix Sec. F. We hope that the new visualization results we added give you more insights into the MAE objective.
>
> **(4) Typo in negative V-A pair**:
>
> Thanks for the catch. We will update x^{V+} to x^{V-} in the final version.

---

### Official Review · Reviewer_yMpw · 2022-07-12

**Rating:** 7
**Confidence:** 4
**Soundness:** 4 excellent
**Presentation:** 4 excellent
**Contribution:** 3 good

**Summary:**

The paper proposes a transformer that takes both speech and images as input to learn a joint vision and language representation. The transformer is trained with two tasks, one is masked autoencoding and the other is whether an image matches an audio clip. The models are then fine tuned on a particular task, for example, audio-to-video retrieval, multimodal sentiment classification, and visual question answering with speech as input. Experiments show that the approach performs well compared to the state of the art.

**Questions:**

> Section 3.2

It is a bit unclear what architecture is used for the decoder. It is also unclear how a decoder could be used to generate both images and spectrogram patches. Images have 3 channels, while a spectrogram only has 1. How is this achieved exactly?

> ... we construct half of the visio-audio pairs inside a batch as mismatched (negative) pairs (x^{V+}, x^A), by replacing video frames x^{V+} ...

Should it be (x^{V-}, x^A)?

> Section 5.3

It's unclear in general how the model is used for each downstream task. Are there any output layers added for each task? Are both the images and spectrograms present during testing? The model in principle should have the ability to take only a single modality instead of both.

**Limitations:**

The societal impact is inherited from the vision-language tasks.

**Strengths And Weaknesses:**

The paper is easy to follow. The proposed architecture is simple, yet performs well. The ablation study and comparison of various input modalities are convincing.

---

> ### Author Response · Authors · 2022-08-02
> **Response to Reviewer yMpw**
>
> Thank you for appreciating the strengths of our paper.
>
> **(1) Decoder architecture**:
>
> We use the single decoder architecture for both visual and audio representations. Then we use two different output layers to map the encoder's last hidden states into visual (3x16x16 patch) and audio (128-dim spectrogram) outputs.
>
> **(2) More details about finetuning on different tasks**:
>
> For each of the downstream tasks, we add a task-specific head on top of the encoder representation.
> - For retrieval tasks, we introduce a two-layer MLP on top of the encoder representation of the [CLS] token to obtain the matching scores $\in$ [0,1], which corresponds to match vs. mismatch pairs, and train the model jointly with binary cross-entropy loss.
> - For visual question answering tasks, we introduce a two-layer MLP on top of the encoder representation of the [CLS] token to obtain the answer probabilities with 3129 answer candidates, and train the model jointly with binary cross-entropy loss in a multi-label classification setup.
> - For multimodal sentiment analysis tasks, we introduce a two-layer MLP on top of the encoder representation of the [CLS] to obtain sentiment scores, and train the model jointly with L2 regression loss.
>
> We will add these details in the final version of the paper.
>
> **(3) Typo in negative V-A pair**:
>
> Thanks for the catch. We will update x^{V+} to x^{V-} in the final version.

---

### Official Review · Reviewer_WYuA · 2022-07-12

**Rating:** 6
**Confidence:** 4
**Soundness:** 2 fair
**Presentation:** 2 fair
**Contribution:** 2 fair

**Summary:**

This paper proposes a vision-language transformer representation learning model, denoted as TVLT. The proposed model is based on masked auto-encoders comprised of two encoders (one for vision and one for audio) and a single shared decoder.
The model was trained to predict the masked segments together with newly proposed Vision-Audio Matching (VAM) contrastive loss. The authors provide results for several tasks namely: Audio-to-Video Retrieval, Multimodal Emotion Analysis, Audio-to-Image Retrieval, and Speech-based Visual Question Answering.

**Questions:**

1) The main issue with this submission is its novelty. By nature audio is pretty different than image, however, the authors used the same encoder arch for both domains where they consider the spectrogram as another image. Did the authors explore other encoders as well?
2) Considering the synthetic speech data, I'm a bit surprised that synthesized speech from the text was more beneficial than just text considering the way this speech was generated. In other words, when speech was synthesized from the text it will contain the content and artificial prosodic feature, and I'm not sure why should it be better than just text? Can the authors provide more details about this process?
3) There are some missing references the authors should pay attention to, especially when considering the textless approach:
* Shi, Bowen, et al. "Learning audio-visual speech representation by masked multimodal cluster prediction." arXiv preprint arXiv:2201.02184 (2022).
* Owens, Andrew, and Alexei A. Efros. "Audio-visual scene analysis with self-supervised multisensory features." Proceedings of the European Conference on Computer Vision (ECCV). 2018.
* Surís, Didac, et al. "Cross-modal embeddings for video and audio retrieval." Proceedings of the European Conference on Computer Vision (ECCV) Workshops. 2018.
* Lakhotia, Kushal, et al. "On generative spoken language modeling from raw audio." Transactions of the Association for Computational Linguistics 9 (2021): 1336-1354.
* Kreuk, Felix, et al. "Textless speech emotion conversion using decomposed and discrete representations." arXiv preprint arXiv:2111.07402 (2021).
* Kharitonov, Eugene, et al. "Text-free prosody-aware generative spoken language modeling." arXiv preprint arXiv:2109.03264 (2021).

**Limitations:**

Yes.

**Strengths And Weaknesses:**

Strengths:
1) The authors provide results for several tasks, demonstrating the effectiveness of the proposed approach.
2) The authors provide a detailed analysis of their method together with an ablation study.

Weaknesses:
1) Novelty is limited. The proposed method is very similar to the masked auto-encoder which was recently proposed for image, where the authors additionally include another audio encoder getting the audio spectrogram "image" as another input.
2) Missing references to previous work.
3) Experimental description can be improved.

---

> ### Author Response · Authors · 2022-08-02
> **Response to Reviewer WYuA**
>
> **Regarding the novelty and contribution of our paper**:
>
> > Novelty is limited. The proposed method is very similar to the masked auto-encoder which was recently proposed for image, where the authors additionally include another audio encoder getting the audio spectrogram "image" as another input.
>
> As explained in L37-L44 of our paper and mentioned by the two other reviewers, our paper's main novelty and contribution are not about inventing novel architectures or the training objectives for existing unimodal tasks. Instead, we focus on demonstrating the possibility of using a minimalist design, where homogeneous transformer blocks process both raw visual and audio input from videos (Sec. 3), and presenting a novel textless vision-and-language (VL) representation learning method based on the design (Sec. 4).
>
> Moreover, we claim that our textless VL model is more compact and efficient than the existing text-based VL models (Sec. 1) and provide empirical validation results (Sec. 6). Reviewer JbWb also mentions such contributions:
> > "achieve comparable results to text-base models while being 50x faster inference"
>
> >  "the minimalist design of TVLT allows to scale up the model so it is expected to give another possibility of the large-scale multi-modal model as GPT-3 did in NLP".
>
> We hope that there can be an increase in your judgment of our paper's academic contribution by considering the above points.
>
>
> **Alternative encoder architecture**:
>
> > By nature audio is pretty different than image, however, the authors used the same encoder arch for both domains where they consider the spectrogram as another image. Did the authors explore other encoders as well?
>
> As explained above, our main goal/contribution is a minimalist design for textless vision-and-language, and hence we used a simple single-encoder approach for this. However, to answer your suggestion and provide empirical evidence for the community, we also experimented with the TVLT architectural variant with a dual/separate encoder. Specifically, similar to CLIP, we use two separate encoders: visual encoder and audio encoder. We initialize both encoder weights with MAE ImageNet checkpoints, following the TVLT joint encoder.
>
> The table below compares the separate encoders with the joint encoder on two tasks: VQA and MSR-VTT. To tackle VQA with separate encoders, we learned a 2-layer self-attention fusion layer over the concatenation of vision and audio encoder hidden states. Our joint encoder architecture achieves better accuracy on both tasks than separate encoder architecture. The results show that although the vision and audio spectrogram are two different modalities, the single joint encoder could learn useful cross-modal representation for VL tasks without needing modality-specific encoders. We hope this addresses your concern about using a single encoder. We will add these architecture ablations in the final version.
>
> | Encoder |    VQA Acc.  |    MSRVTT R@1 |
> | --- | --- | ---  |
> | Seperate  |         53.1                  |    9.6 |
> | Joint (current)         |       **54.6**                   |  **10.2** |
>
> **Synthesized speech was more beneficial than text**:
>
> As described in L213-214 and L222-223, we use TTS only for the VQA task, where text questions achieve *higher* accuracy than TTS-based audio questions. Therefore, we did **not** conclude that synthesized speech was better than text.
> Although Table 1 in the main paper shows results on audio-to-video retrieval and multimodal sentiment analysis, where speech-based language representation was more effective than text, please note that we use real audio from video, instead of synthesized speech in these experiments.
> As described in L267-269, we believe that acoustic features such as tone and loudness from real audio, (not synthesized audio) could be helpful for some tasks such as emotion analysis.
>
> **Missing references**:
>
> We appreciate the suggestions on the references for audio input, and we will incorporate them in the final version of the paper. We still would like to emphasize that although some of these works share the idea of “textless” with ours via audio inputs, the focus of our work is different from these works - we demonstrate the possibility of using a minimalist design, where homogeneous transformer blocks process both raw visual and audio input from videos (Sec. 3), and present a novel textless vision-and-language (VL) representation learning method based on the design (Sec. 4).

---

> > ### Comment · Reviewer_WYuA · 2022-08-08
> > **Response to authors**
> >
> > I would like to thank the authors for the additional results and clarifications.
> > After reading the author's response, I understand the author's claims and agree that the simplicity of the arch. is a big advantage. Hence, I'm increasing my score to 6.
> >
> > I do have a few clarifications
> > 1) **Regarding the encoders:** Thanks for the additional results. However, I was referring to actual architecture. Meaning did the authors try using different model architecture for speech and image? or only the same architecture but separate encoders?
> > 2) **Paper update:** I highly encourage the authors to update the paper to include the new modifications before the author's discussion ends.

---

> > > ### Author Response · Authors · 2022-08-08
> > > **Response to Reviewer WYuA**
> > >
> > > Thanks for appreciating our claims and increasing the score!
> > >
> > > > 1. Regarding the encoders: Thanks for the additional results. However, I was referring to actual architecture. Meaning did the authors try using different model architecture for speech and image? or only the same architecture but separate encoders?
> > >
> > > For the additional experiment, we use separate vision/audio encoders with the same homogeneous transformer architecture.
> > > As previous work showed the usefulness of a transformer encoder for unimodal vision or audio inputs (L95-113), in this work, we explored the possibility of using a single joint transformer encoder for cross-modal representation learning.
> > > We did not use different architectures for separate encoders.
> > > The only difference in input encoding of vision/audio modalities lies in different input patch embeddings to the encoders, where we encode an image (or video frame) as a sequence of 16x16-sized patches and encode a spectrogram as a sequence of 2x128 (time x frequency)-sized patches.
> > >
> > > > 2. Paper update: I highly encourage the authors to update the paper to include the new modifications before the author's discussion ends.
> > >
> > > Thanks, we also would like to update our paper to include the new modifications.
> > > However, the NeurIPS 2022 FAQ for Authors (https://neurips.cc/Conferences/2022/PaperInformation/NeurIPS-FAQ), mentions that *"Please stick to the 9-page limit during the review process. The extra page applies only to the accepted papers for the camera-ready."*
> > > Thus, we updated the modifications in the appendix in the supplementary files and colored the updated contents in blue fonts. We will update the modifications later in the camera-ready version and move them to the main paper’s allowed 10th page.

---

### Meta-Review · Area_Chair_metX · 2022-08-26

**Recommendation:** Accept
**Confidence:** Certain

**Metareview:**

Reviewers acknowledge that the proposed method is simple and performs well and have the potential to present an interesting research direction for future works. The authors respond actively to the review comments and one reviewer raises the score after author response. It is a good paper, and I recommend acceptance. Authors should follow the reviewer's suggestions to update the paper to address some the questions.

**Award:**

No

---

### Decision · Program_Chairs · 2022-09-14

Accept